# Peer review of "Coordinated Regulation of Myeloid-Derived Suppressor Cells by Cytokines and Chemokines"

_cancers, 2022, doi:10.3390/cancers14051236_

Round 1

Reviewer 1 Report

Myeloid‐derived suppressor cells (MDSCs) are immature myeloid cells that impair immune cell functions and promote tumor progression. More and more evidence indicates that cytokines and chemokines in the tumor microenvironment regulate MDSCs. In this review, the authors summarize the current evidence on the regulation of myeloid‐derived suppressor cells by cytokines and chemokines. The review is well written. It is clear and succinct. Some minor revisions are suggested before publication:

  1. The authors discussed the cytokines/chemokines regulating neutrophils and monocytes in cancer. They can emphasize a little more about the reason behind choosing neutrophils and monocytes over other immune cells.
  2. The authors summarized some literature about cytokines/chemokines regulating monocytes, which should be extended into more detail.
  3. Some grammar errors.

Reviewer 2 Report

The review paper by Ru Li and colleagues, is a well-written, comprehensive and updated overview of one of the most recent research breakthroughs in the field of cancer immunotherapy. I read the manuscript with interest. I have a few concerns though:

1. The part of characteristic features of MDSCs could further improve. I suggest giving more emphasis on markers used to identify MDSC populations.

2. The results of recent studies using the MDSC and their signalling pathway could be present in a more detailed way and maybe it would be a good idea to add a summary table.

Round 2

Reviewer 2 Report

Authors responded to all my suggestions. I have no further comments.